# Antimicrobial Susceptibility and Characterization of Resistance Mechanisms of *Corynebacterium*
*urealyticum* Clinical Isolates

**DOI:** 10.3390/antibiotics9070404

**Published:** 2020-07-13

**Authors:** Itziar Chapartegui-González, Marta Fernández-Martínez, Ana Rodríguez-Fernández, Danilo J. P. Rocha, Eric R. G. R. Aguiar, Luis G. C. Pacheco, José Ramos-Vivas, Jorge Calvo, Luis Martínez-Martínez, Jesús Navas

**Affiliations:** 1Grupo BIOMEDAGE, Facultad de Medicina, Universidad de Cantabria, Herrera Oria 2, 39011 Santander, Spain; ichapartegui@idival.org; 2Instituto de Investigación Valdecilla (IDIVAL), Herrera Oria s/n, 39011 Santander, Spain; mfernandez@idival.org (M.F.-M.); jvivas@idival.org (J.R.-V.); jorge.calvo@scsalud.es (J.C.); 3Servicio de Microbiología, Hospital Universitario Marqués de Valdecilla (HUMV), Av. Valdecilla, 25, 39008 Santander, Spain; ana.rodriguez@scsalud.es; 4Instituto de Ciências da Saúde, Universidade Federal da Bahia, Av. Reitor Miguel Calmon, s/n —Canela, Salvador 40231-300, Brasil; djobimpassos@gmail.com (D.J.P.R.); lgcpacheco@gmail.com (L.G.C.P.); 5Department of Biological Sciences, Center of Biotechnology and Genetics, State University of Santa Cruz (UESC), Ilhéus 45662-90, Brazil; ericgdp@gmail.com; 6Unidad de Gestión Clínica, Hospital Universitario Reina Sofía, Av. Menéndez Pidal, s/n, 14004 Córdoba, Spain; luis.martinez.martinez.sspa@juntadeandalucia.es; 7Departamento de Microbiología, Universidad de Córdoba, Campus Rabanales. Edif. Severo Ochoa (C6), 14071 Córdoba, Spain; 8Instituto Maimónides de Investigación Biomédica de Córdoba (IMIBIC), Av. Menéndez Pidal, s/n, 14004 Córdoba, Spain

**Keywords:** *Corynebacterium urealyticum*, multidrug resistance, coryneform, antimicrobials, whole genome sequencing

## Abstract

*Corynebacterium urealyticum* is a non-diphtherial urease-producing clinically relevant corynebacterial, most frequently involved in urinary tract infections. Most of the *C. urealyticum* clinical isolates are frequently resistant to several antibiotics. We investigated the susceptibility of 40 *C. urealyticum* isolated in our institution during the period 2005–2017 to eight compounds representative of the main clinically relevant classes of antimicrobial agents. Antimicrobial susceptibility was determined by the Epsilometer test. Resistance genes were searched by PCR. All strains were susceptible to vancomycin whereas linezolid and rifampicin also showed good activity (MICs_90_ = 1 and 0.4 mg/L, respectively). Almost all isolates (39/40, 97.5%) were multidrug resistant. The highest resistance rate was observed for ampicillin (100%), followed by erythromycin (95%) and levofloxacin (95%). Ampicillin resistance was associated with the presence of the *blaA* gene, encoding a class A β-lactamase. The two rifampicin-resistant strains showed point mutations driving amino acid replacements in conserved residues of RNA polymerase subunit β (RpoB). Tetracycline resistance was due to an efflux-mediated mechanism. Thirty-nine PFGE patterns were identified among the 40 *C. urealyticum*, indicating that they were not clonally related, but producing sporadic infections. These findings raise the need of maintaining surveillance strategies among this multidrug resistant pathogen.

## 1. Introduction

The genus *Corynebacterium* include Gram positive aerobic bacteria, which are widely distributed in the microbiota of humans and animals. Medically relevant *Corynebacterium* species consist of *Corynebacterium diphtheriae*, the pathogenic bacterium that causes diphtheria, and the non-diphtherial corynebacterial (as *Corynebacterium urealyticum*, *Corynebacterium striatum*, and *Corynebacterium jeikeium*, among others), which are part of the skin and mucous membranes flora [1]. They are usually not pathogenic but can occasionally opportunistically capitalize on atypical access to tissues or weakened host defenses. A key role of *Corynebacterium* species as attenuator of *Staphylococcus aureus* virulence in the nose microbiota has been recently suggested [2]. *Corynebacterium urealyticum* is a slow growing, asaccharolytic, and lipophilic microorganism, whose name refers its ability to split urea [3]. It possesses a strong urease activity that leads to the formation of struvite stones following ammonium magnesium phosphate precipitation due to the increase of urine pH. *C. urealyticum* behaves as an opportunistic human pathogen, causing acute and chronic urinary tract infections (UTIs), eventually leading to bacteraemia [3]. It has also been isolated from the skin of healthy elderly individuals, mainly females [4]. Some of the risk factors pre-disposing to an infection by *C. urealyticum* are prolonged use of urinary catheters, hospitalization for long periods, previous treatment with broad-spectrum antibiotics or immunosuppressants, and history of previous UTIs [5].

The majority of *C. urealyticum* currently isolated from clinical samples are multidrug resistant, thus potentially limiting effective empirical treatment [1,6]. Development of resistance has been observed during treatment with different antimicrobial classes: β-lactams, gentamicin, fluoroquinolones, macrolides, rifampicin, and tetracycline [7]. However, the resistance mechanisms for most of these compounds have not been described.

Previous [8,9] and recent studies [10] about antimicrobial activity against *C. urealyticum* and other *Corynebacterium* species have proven that vancomycin and linezolid were uniformly active against these bacteria, whereas most of them displayed high level resistance against quinolones, β-lactams, and macrolides.

The aim of this study was to evaluate the prevalence of multidrug resistant strains as well as determine the resistance mechanisms of *C. urealyticum* isolates from a Spanish hospital (Santander) during the period 2005–2017. The genomes of five of these isolates have been sequenced and the main antimicrobial resistance determinants identified.

## 2. Results

### 2.1. Susceptibility of C. urealyticum to Antimicrobial Agents

The MIC_50_ and MIC_90_ distributions, as well as the percentage of resistance to the different antibiotics for the 40 *C. urealyticum* included in this study, are shown in Table 1.

Ampicillin (100%), erythromycin (95%), and levofloxacin (95%) showed the highest number of resistant isolates, with a monomodal distribution of their MICs. Interestingly, for each antimicrobial agent, the two susceptible strains were different (VH4696 and VH4851 susceptible to erythromycin, VH6223 and VH4248 susceptible to levofloxacin). All the levofloxacin and erythromycin-resistant isolates had high level resistance (MIC > 32 mg/L and MIC > 256 mg/L, respectively). Concerning ampicillin-resistant isolates, 39 isolates showed a MIC > 256 mg/L whereas one isolate showed a MIC = 24 mg/L (the urine isolate VH2234).

Gentamicin and tetracycline also showed high number of resistant *C. urealyticum* strains, with 82.5% and 50% of isolates, respectively. In both cases, a bimodal MICs distribution was observed. The MICs values for gentamicin were in the range 1.5–256 mg/L, and between 3 and 256 mg/L for tetracycline, with relatively low MIC_90_ values, 9 and 4 mg/L, respectively.

Rifampicin and linezolid were the compounds with the lowest percentage of resistance, with only two and one resistant strains, respectively. Rifampicin-resistant isolates showed a high level resistance (MICs > 32 mg/L), whereas the MIC of the linezolid resistant isolate was 3 mg/L. Vancomycin was the only compound uniformly active against all tested isolates.

On the whole, multidrug resistance, defined as nonsusceptibility to at least one agent in three or more antimicrobial categories [11], was observed in 39 out of 40 isolates.

All the strains were resistant to at least two antimicrobial compounds. They presented nine different resistance profiles, the resistance combination for levofloxacin (LVX), ampicillin (AMP), gentamicin (GEN) and erythromycin (ERY) being the most prevalent (LVX-AMP-GEN-ERY = 14 isolates), as well as in combination with tetracycline (TET) resistance (LVX-AMP-GEN-ERY-TET = 13 isolates). The relationship between antibiotic resistance profiles and sample origin could not be established.

Strain VH4549, isolated from a urine sample, was resistant against six of the eight tested compounds (it showed LVX-AMP-GEN-ERY-TET-RIF profile), being the isolate with less therapeutic options. On the other hand, the strain VH6223, from placenta, was the most susceptible isolate, showing resistance only against ampicillin and erythromycin.

The susceptibility of the five *C. urealyticum* whose genomes were sequenced against the eight compounds tested is shown in Table 2. 

### 2.2. Detection of Resistance Genes by PCR and Genome Sequencing

The 38 *C. urealyticum* resistant to erythromycin carried the *ermX* gene. The PCR amplification product of strain VH2234 was sequenced and compared with the *ermX* gene of *C. urealyticum* DSM 7109 [4], showing a 95% identity. Whole genome sequencing of five erythromycin-resistant *C. urealyticum* confirmed the presence of the *ermX* gene.

Thirty-eight out of 40 *C. urealyticum* were resistant to levofloxacin. The 38 resistant strains showed a MIC of levofloxacin >32 mg/L. The sequences of the QRDR region of the *gyrA* gene of 28 isolates categorized as resistant and one isolate categorized as susceptible were compared to the sequence of this region in the *gyrA* gene of *C. urealyticum* DSM 7109 (quinolone-susceptible). Twenty-two levofloxacin-resistant isolates showed the double mutation Ser-90→Val and Asp-94→Tyr, whereas in three resistant strains Asp-94 was replaced by Ala. Three levofloxacin-resistant strains were single Ser-90→Val mutants. The levofloxacin susceptible strain VH4248 did not show mutations at residues 90 and 94, as the reference strain DSM 7109. One strain resistant to levofloxacin showed no mutation at the QRDR region of *gyrA*, suggesting a different resistance mechanism.

Two of our isolates (VH3073 and VH4549), as well as *C. urealyticum* DSM 7109, were resistant to rifampicin (MIC > 32 mg/L). Rifampicin resistance is nearly always due to a genetic change in the β subunit of RNA polymerase (RpoB). Alignment of the RpoB sequences of these three rifampicin-resistant strains with the corresponding proteins of three rifampicin-susceptible *C. urealyticum* revealed non-conservative changes in Ser-444 (VH4549 and DSM 7109) or Gln-511 (VH3073), which can be related with the rifampicin-resistant phenotype (Figure 1).

The 40 *C. urealyticum* strains were ampicillin-resistant. The high ampicillin resistance phenotype (MIC_90_ > 256 mg/L) was associated with the presence of the *blaA* gene. Ampicillin-resistant *C. urealyticum* rendered the expected 0.8 Kb PCR product when amplified with *blaA* specific primers. Conversely, *C. urealyticum* 18408721, a clinical isolate susceptible to ampicillin, was negative by the *blaA*-based PCR. The *C. urealyticum blaA* gene encodes a serine hydrolase belonging to the class A β-lactamase protein family. In order to know the genomic context of the resistance genes and inquiry about their transfer mechanisms, we sequenced the genomes of five *C. urealyticum*. Genome analysis revealed that the region containing the *blaA* gene is highly conserved among the five *C. urealyticum* and the previously sequenced strains *C. urealyticum* DSM 7109 [4] and DSM 7111 [12] (Figure 2A). This region spans 20 Kbp of the assembled genomes, including a *tnp* gene (transposase), *lysR* (transcriptional regulator), and a Penicillin-Binding-Protein (PBP) type 1 gene. A similar genomic organization can be found in *C. striatum* KC-Na-01 (Figure 2B). We compared the amino acid sequence encoded by the *C. urealyticum blaA* gene with its counterparts in other species and we found that it is highly conserved in *C. striatum* strain KC-Na-01 (NCBI’s protein accession #WP_049063072), in *C. jeikeium* K411 (#WP_034987125), in *C. amycolatum* SK46 (#WP_076773763.1), as well as in *C. resistens* DSM 45100 (#WP_042378726.1) [13]. There is a particular region in the BlaA sequence (between amino acid positions 157–164) that concentrates most of the variability when comparing all species.

Thirty-three of the *C. urealyticum* showed low level gentamicin resistance. The presence of the gene *aac(3)-XI* encoding for an aminoglycoside 3-N acetyltransferase was evaluated by PCR using primers based on the *C. striatum aac(3)-XI* gene [14], giving negative results in all strains. However, analysis of the five *C. urealyticum* sequenced genomes revealed the presence of a 447-bp open reading frame showing 79% identity with the *C. striatum aac(3)-XI* gene in four strains but not in strain VH4248 (Figure 3). Homology search in databases revealed that the *C. urealyticum aac(3)-XI* orthologous encodes an aminoglycoside 3-N acetyltransferase also present in *C. coyleae* (#WP_092102070.1) (80% identity), *C. fournierii* (#WP_085957501) (75% identity), and several *Corynebacterium* spp. The *aac(3)-XI* gene is flanked by *arfB*, encoding the peptidyl-tRNA hydrolase ArfB, at the upstream region, and the *luxR*-family two-component transcriptional response regulator, at the downstream region (Figure 3). On the other hand, a search for additional aminoglycoside resistance genes revealed the presence of the gene *aph(3′)-Ic*, encoding resistance to kanamycin and other aminoglycosides rarely used in clinical practice, and the pair of genes *aph(3″)-Ib* and *aph(6)-Id*, conferring streptomycin resistance, in four strains, but not in strain VH4248, which does not present any of these genes.

Twenty of our *C. urealyticum* were resistant to tetracycline. Tetracycline-resistant strains showed a bimodal MICs distribution: strains VH638 and VH2234 showed high resistance level (MICs ≥ 256 and 32 mg/L, respectively), whereas 18 strains showed low resistance level (MICs range = 3–6 mg/L). The *C. urealyticum* reference strain DSM 7109 is tetracycline-resistant (MIC = 32 mg/L) and this resistance is associated to the *tetAB* genes [4]. The *tetAB* genes were neither detected by genome analysis of the strains VH4549, VH5757, and VH5913, nor by PCR analysis of the remaining 17 tetracycline-resistant strains, which suggests the existence of alternative resistance mechanisms. When the tetracycline MICs of seven of our tetracycline-resistant strains were measured in the presence of the efflux-pump inhibitor Phe-Arg-β-naphthylamide (PAβN), a dramatic increase of tetracycline susceptibility was observed (Table 3). However, the tetracycline MIC of strain DSM 7109 remained unchanged. 

### 2.3. Molecular Epidemiology of the C. urealyticum Isolates

The PFGE method displayed a high typeability and discriminatory power. *XbaI* digestion of the 40 *C. urealyticum* isolates revealed 39 distinct PFGE patterns which were labelled from 1 to 39 (Figure 4). The reference strain DSM 7109 PFGE pattern was also included in the dendrogram. Only two strains were assigned into the same PFGE pattern (VH4696 and VH4851), designated as pattern 18. These two strains were isolated from urine samples taken from the same patient at an interval of eight days, and showed the same antibiotic resistance profile (LVX-AMP-GEN-TET). The relationship between PFGE patterns and antibiotic resistance profiles or sample origin could not be established.

The high number of pulsotypes obtained in our *C. urealyticum* isolates highlighted the elevated genetic diversity in this specie. Among 40 strains, 39 were unrelated, producing sporadic infections.

## 3. Discussion

Management of *C. urealyticum* infections usually relies on glycopeptides (vancomycin or teicoplanin), to which this microorganism is uniformly susceptible [15]. All our *C. urealyticum* were susceptible to vancomycin. In two previous studies, we have reported full activity of vancomycin [16] and teicoplanin [17] against *C. urealyticum*. In a more recent report including 52 *C. urealyticum* clinical strains, vancomycin was also active against all the isolates [10]. In vitro studies indicated that linezolid can also be effective [5], and subsequent studies confirmed that linezolid was fully active against *C. urealyticum* [8,9,10,18]. The data presented in this work basically agree with linezolid susceptibility data from the mentioned studies, highlighting its option as a therapeutic alternative for vancomycin. However, one of our isolates showed low level resistance to this compound (MIC = 3 mg/L), which raises the need of maintaining surveillance strategies among this multidrug resistant pathogen as well as defining its resistance profile before treatment.

Isolates resistant against erythromycin, levofloxacin, and ampicillin showed a monomodal distribution of their MICs, which suggests the existence of a unique or major mechanism of resistance for each antimicrobial. Erythromycin was inactive against the majority of our *C. urealyticum*, in accordance with previously reported data [19]. All the erythromycin-resistant *C. urealyticum* carried the *ermX* gene. It is well established that the *ermX* gene encodes an N-6-methyltransferase that modifies an adenine of the 23S rRNA, conferring resistance against erythromycin [4]. Whole genome sequencing of five *C. urealyticum* confirmed the presence of the *ermX* gene.

Fluoroquinolones have been extensively used in the empirical treatment of urinary tract infections. Upon antibiotic administration, these drugs tend to accumulate in the organs of the body leading to the selection of spontaneous mutants in large bacterial populations, including those that colonize the skin and mucous membranes such as corynebacteria. In fluoroquinolone-resistant *Corynebacterium* spp. mutations are circumscribed to the *gyrA* gene (QRDR region), since these bacteria lack the *parC* gene. Thirty-eight of our 40 *C. urealyticum* (95%) showed high level resistance to levofloxacin. López-Medrano et al. reported that 79% of their isolates were resistant to ciprofloxacin [18]. Sequencing of the QRDR region of the *gyrA* gene of 29 levofloxacin-resistant *C. urealyticum* revealed that resistance is associated with single or double amino acid substitutions in residues Ser-90 and Asp-94 (*C. urealyticum* numbering). Ramos et al. have recently reported three *C. urealyticum* strains showing high level quinolone resistance associated to double amino-acid substitutions in Ser-90 and Asp-94 [20]. However, in three of our strains, one amino acid replacement (Asp-94 by Tyr) was enough to display high resistance level to levofloxacin. In one *C. urealyticum*, no link between mutations in this region and levofloxacin-resistant phenotype could be established, suggesting the existence of additional resistance mechanisms.

Rifampicin has been used as complementary agent for the management of *C. striatum* infections [21,22]. Rifampicin showed good activity against our *C. urealyticum*, since only two out of 40 isolates were resistant. Of note, the *C. urealyticum* reference strain (DSM 7109) is also rifampicin-resistant. A recent study including 52 *C. urealyticum* in Canada showed the same MIC_50_ and MIC_90_ values for rifampicin (≤0.05 mg/L) [10]. Resistance to this compound typically results from the substitution of some highly conserved residues in the RNA polymerase β subunit [23]. In *Mycobacterium tuberculosis*, more than 96% of rifampicin-resistant strains have mutations within the 81-bp rifampicin resistance-determining region (RRDR) of the *rpoB* gene (codons 507–533) [24]. We compared RpoB sequences of the three rifampicin-resistant *C. urealyticum* with that of susceptible strains. Considering the presumptive location of the RpoB active site and discarding the influence of conservative replacements in rifampicin susceptibility, we propose that the high rifampicin MICs can be explained by amino acid replacements in RpoB of the two rifampicin-resistant strains (Ser-444→Phe in VH4549 and Gln-511→Lys in VH3073) (Figure 1). We have also identified the substitution Ser-444→Asn in DSM 7109.

All our *C. urealyticum* displayed high level resistance to ampicillin (MIC_90_ > 256 mg/L). Hydrolysis of β-lactam antibiotics by β-lactamases is the most common mechanism of β-lactam resistance in clinically relevant bacteria. The ampicillin-resistant *C. urealyticum* were positive for the *blaA*-based PCR whereas a susceptible strain was negative. Whole genome analysis of five *C. urealyticum* isolates confirmed the presence of the *blaA* gene, flanked by transposase encoding genes (Figure 2A). The *blaA* gene encodes a serine hydrolase belonging to the class A β-lactamase protein family, which is highly conserved in several *Corynebacterium* species. In seven *C. urealyticum* strains and in *C. striatum* KC-Na-01 the *blaA* gene is in close vicinity to a transposase encoding genes (Figure 2B), suggesting that it has been horizontally propagated.

Aminoglycosides are complementary antibiotics for the treatment of infections caused by *Corynebacterium* spp. However, it has been reported that *C. urealyticum* is mostly resistant to aminoglycosides [25]. The MICs of *C. urealyticum* DSM 7109 for kanamycin and streptomycin are >256 and >128 mg/L, respectively [4]. Thirty-three out of our 40 *C. urealyticum* were resistant to gentamicin, with MIC_50_ and MIC_90_ values of 4 and 9 mg/L, respectively, indicating a high prevalence but a low level of resistance (range tested 0.016–256 mg/L). The gene *aac(3)-XI*, encoding an aminoglycoside 3-N acetyltransferase, which confers resistance to gentamicin and other aminoglycosides, has been recently identified in *C. striatum* [14], whose presence was correlated to low level of resistance to gentamicin [26]. Search by PCR with primers based on *C. striatum aac(3)-XI* sequence gave negative results in our 40 *C. urealyticum*, since these primers did not match with the *C. urealyticum aac(3)-XI* gene. However, by means of whole genome sequencing, we detected this gene as part of a highly conserved region in strains VH3073, VH4549, VH5757, and VH5913, but not in strain VH4248. We hypothesize that low level gentamicin resistance in these four strains is related with the presence of the *aac(3)-XI* gene, whereas in strain VH4248 is due to another mechanism. This analysis also revealed the presence in these four strains of a region including the gene *aph(3′)-Ic* (related to kanamycin resistance) and the pair of genes *aph(3″)-Ib* and *aph(6)-Id* (related to streptomycin resistance), which is also present in *C. urealyticum* DSM 7109 [4] as well as in other *Corynebacterium* spp. 

Fifty percent of our *C. urealyticum* were resistant to tetracycline. Tetracycline-resistant strains showed a bimodal MICs distribution, which suggests the existence of different resistance mechanisms. In *C. striatum*, tetracycline resistance is mediated by an ATP gradient efflux mechanism encoded by the pair of genes *tetA-tetB* [27], which is also found in *C. urealyticum* DSM 7109 [4]. However, in our *C. urealyticum*, the *tetA-tetB* genes were not detected. The remarkable decrease of tetracycline MICs of seven of our *C. urealyticum* observed in presence of PAβN, a broad-spectrum efflux pump inhibitor, indicates that tetracycline resistance is mediated by an efflux mechanism. 

PFGE is considered as the “gold standard” technique to assess epidemiological relationships for most clinically-relevant bacteria [28]. Our results showed almost the same number of isolates as PFGE patterns, with only two strains sharing the same pulsotype. This high diversity among *C. urealyticum* isolates revealed that they are not related but causing sporadic infections. While whole genome sequencing provides more detailed and accurate information, its use is not still viable in the daily clinical routine. Thus, PFGE remains as an important tool at epidemiological level. However, increasing the number of sequenced strains will provide more information, such as antimicrobial resistance and virulence, in comparison to PFGE, which will improve, in turn, our knowledge about the resistance and virulence mechanisms of this pathogen.

## 4. Materials and Methods 

### 4.1. Bacterial Strains and Growth Conditions

Forty *C. urealyticum* isolated from clinical samples at Clinical Microbiology Laboratory, Hospital Universitario Marqués de Valdecilla (HUMV), Santander (Spain), during the period 2005–2017, were used in this study. *C. urealyticum* DSM 7109 was also included as the reference strain. The origin of the samples was diverse: urine (25), abdominal drainage (3), surgical wound (3), blood (3), skin ulcer (2), urinary stone (1), non-surgical wound (1), diabetic foot ulcer (1), and placenta (1). They were initially identified by the API Coryne system (bioMérieux, Marcy l’Etoile, France) and confirmed by MALDI-TOF mass spectrometry using the Vitek MS (bioMérieux) platform, according to manufacturer’s instructions. *C. urealyticum* 18408721, isolated at Hospital Universitario Central de Asturias (HUCA), Oviedo (Spain), was used as an ampicillin-susceptible control strain. All strains were grown in blood agar (BA) plates at 37 °C for 72 h and kept frozen at −80 °C in Brain Heart Infusion (BHI) broth with 20% glycerol until use.

### 4.2. Antimicrobial Susceptibility Assays

To study the activity of eight antimicrobial compounds (ampicillin, erythromycin, gentamicin, levofloxacin, linezolid, rifampicin, tetracycline, and vancomycin) against the 40 *C. urealyticum*, minimal inhibitory concentrations (MICs) were determined using Etest® strips (bioMérieux) on Mueller-Hinton (MH) agar plates supplemented with horse blood and β-NAD (Oxoid, Madrid, Spain). Briefly, agar plates were inoculated with a 100 μL aliquot of a bacterial suspension at OD_600_ = 0.1, and incubated for 48 h. Tetracycline MICs were also determined in presence of PAβN (50 mg/L). 

Clinical categories were established according to the breakpoints for the microdilution susceptibility assay defined by the European Committee on Antimicrobial Susceptibility Testing (EUCAST) (http://www.eucast.org) guidelines [29]. EUCAST defined *Corynebacterium* spp. specific breakpoints for vancomycin, linezolid, tetracycline, and rifampicin; for ampicillin, gentamicin, and levofloxacin we considered EUCAST PK-PD breakpoints. For erythromycin, *Staphylococcus* spp. cut-offs defined by EUCAST were used.

### 4.3. Search of Resistance Genes by PCR

The 40 *C. urealyticum* were screened for the presence of resistance genes commonly found in *Corynebacterium* spp. and other Gram positive multidrug resistant bacteria. Specific primers are listed in Table 4. The *gyrA* and *rpoB* genes were amplified and sequenced for mapping mutations associated with levofloxacin and rifampicin resistance, respectively. *C. urealyticum* DNA for PCR reactions was prepared using InstaGene^TM^ Matrix (Bio-Rad, Madrid, Spain) following manufacturer’s instructions.

### 4.4. PCR Products Sequencing

PCR products were purified with silica gel columns using NucleoSpin® Gel and PCR clean-up kit (Macherey-Nagel, Düren, Germany). Purified DNA was sequenced by Macrogen (Madrid, Spain) with the primers outlined in Table 4. Mutations in *gyrA* and *rpoB* genes and amino acid changes in their corresponding proteins were identified by pairwise and multiple alignment of sequences between resistant and susceptible isolates using MEGA7 [33] and Clustal W [34] programs.

### 4.5. Genome Sequencing and Analysis

For whole genome sequencing, genomic DNAs of strains VH3073, VH4248, VH4549, VH5757, and VH5913 (selected on the basis of their resistance profile), were extracted using the NucleoSpin® Microbial DNA kit (Macherey-Nagel). Library preparation followed the NEBNext Fast DNA Fragmentation and Library Preparation Kit (New England Biolabs, Beverly, MA, USA) protocol and sequencing was performed in an Illumina HiSeq 2500 machine, at above 1000× coverage for all strains. De novo genome assembly was done with the SPAdes assembler and built-in on the PATRIC assembly server [35]. Structural and functional annotations were performed in RAST server (https://rast.nmpdr.org) [36]. Prediction of antimicrobial resistance profiles and comparative genomic analyses were performed using the bioinformatic platforms PATRIC (https://www.patricbrc.org/) [35] and EDGAR [37]. Multiple alignments were performed using T-coffee server in its variant M-coffee [38].

### 4.6. Pulsed-Field Gel Electrophoresis (PFGE)

PFGE was performed with a CHEF-DRIII system (Bio-Rad). Bacteria were grown in BHI broth with shaking at 37 °C for 48−72 h. Cultures were adjusted to OD_600_ = 2.0, cells from 250 μL were pelleted and resuspended in 300 μL of TE buffer (10 mM Tris, 1 mM EDTA) containing 2 mg/mL lysozyme. This suspension was incubated at 37 °C for 1 h, inverting the tubes every 10 min. An equal volume of 2% LM agarose (Pronadisa, Madrid, Spain) in TE buffer containing 1% SDS and 0.2 mg/mL proteinase K was added, and plugs were cast with a standard casting tray. After the plugs solidified, they were incubated overnight at 55 °C with shaking in 4 mL of TE buffer containing 1% sarcosyl and 0.15 mg/mL proteinase K. The plugs were washed six times with pre-warmed TE buffer and then digested with 30 U of *XbaI* at 37 °C overnight. Electrophoresis was performed in a 1.2% agarose gel at 6 V/cm and at 14 °C with 0.5× TBE buffer (0.5 mM Tris, 45 mM boric acid, 0.5 mM EDTA). Pulse times ramped from 0.1 to 5 s for 24 h. Low range PFGE marker (New England Biolabs) was used as the molecular size marker.

Cluster analysis was performed with Fingerprinting II v4.5 software (Bio-Rad) by using the Dice similarity coefficient and the Unweighted Pair Group Method with Arithmetic means (UPGMA), with 1.3% of optimization and tolerance. Isolates were classified as indistinguishable if they showed 100% similarity, as closely related subtypes if they showed 95–99% similarity, and as different strains if they showed <95% similarity.

### 4.7. Data Availability

The whole genomes of the five *C. urealyticum* have been deposited in the Integrated Microbial Genomes Database under the following accession numbers: VH3073 (2833973259); VH4248 (2833948465); VH4549 (2833950470); VH5757 (2833952634); VH5913 (2830819286). Four of them have also been deposited in GenBank under the following accession numbers: VH3073: GCA_008244525.1; VH4248: GCA_008180085.1; VH4549: GCA_008180045.1; VH5913: GCA_008180065.1.

## 5. Conclusions

This study illustrates the high prevalence of multidrug resistance among the *C. urealyticum* isolated in a Spanish hospital, in particular resistances to ampicillin, erythromycin, and levofloxacin. Our isolates were still susceptible to low concentrations of vancomycin and linezolid. In any case, appropriate antimicrobial therapy must be given in accordance with the results of the antimicrobial susceptibility test for each infection. One *C. urealyticum* showed resistance to linezolid. This finding raises the need of maintaining surveillance strategies among this multidrug resistant pathogen.

## Figures and Tables

**Figure 1 antibiotics-09-00404-f001:**
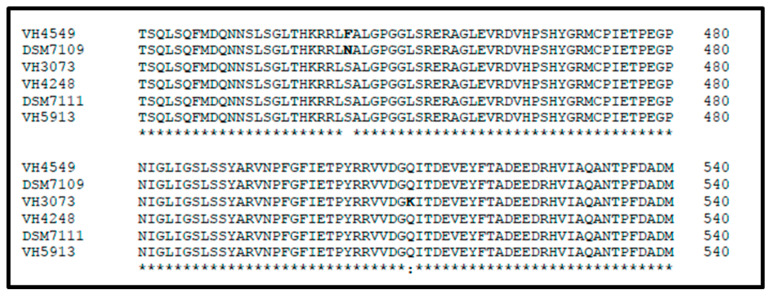
Alignment of RNA polymerase β subunit (RpoB) (residues 421–540) of six *Corynebacterium urealyticum* strains. Strains VH4549, DSM 7109, and VH3073 are resistant to rifampicin whereas VH4248, DSM 7111, and VH5913 are susceptible.

**Figure 2 antibiotics-09-00404-f002:**
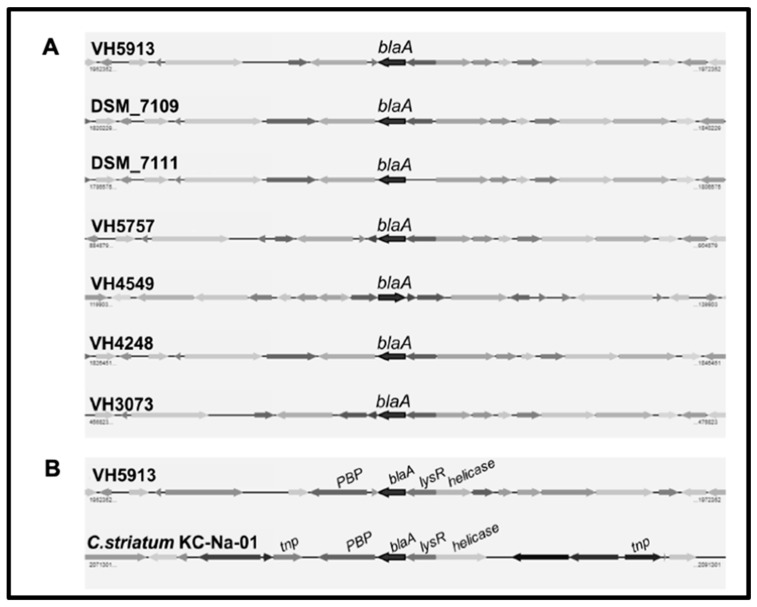
**A**: Genomic context of the *blaA* gene in seven *C. urealyticum*. **B**: Comparison among the *blaA*-containing regions of *C. urealyticum* VH5913 and *Corynebacterium striatum* KC-Na-01.

**Figure 3 antibiotics-09-00404-f003:**
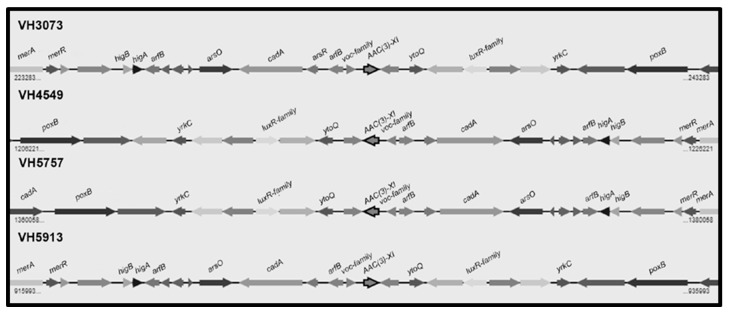
Genomic map of the region including the *aac(3)-XI* gene and its neighbors in four *C. urealyticum* clinical isolates: *merA* (mercuric ion reductase); *merR* (transcriptional regulator, MerR family); *higB* (toxin HigB); *higA* (antitoxin HigA); *arfB* (peptidyl tRNA hydrolase ArfB); *arsO* (flavin-dependent monooxygenase ArsO, associated with arsenic resistance); *cadA* (copper-translocating P-type ATPase); *arsR* (transcriptional regulator, ArsR family); *voc* family (VOC family protein); *ytoQ* (uncharacterized protein YtoQ); *luxR*-family (two-component transcriptional response regulator, LuxR family); *yrkC* (uncharacterized protein YrkC); *poxB* (pyruvate dehydrogenase).

**Figure 4 antibiotics-09-00404-f004:**
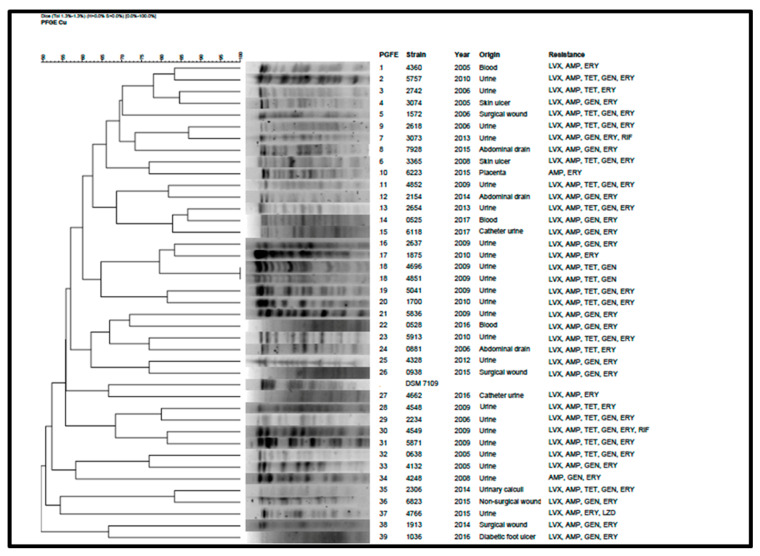
PFGE patterns, dendrogram, sample origin, and resistance profiles of the 40 *C. urealyticum* and the reference strain DSM 7109. LVX: levofloxacin; AMP: ampicillin; TET: tetracycline; GEN: gentamicin; ERY: erythromycin; RIF: rifampicin; LZD: linezolid.

**Table 1 antibiotics-09-00404-t001:** Susceptibility of 40 *Corynebacterium urealyticum* clinical isolates against eight antimicrobial agents. Isolates were classified as resistant or susceptible according to criteria defined by EUCAST (2020). MIC, minimum inhibitory concentration; MIC_50_, MIC that inhibits 50% of the isolates; MIC_90_, MIC that inhibits 90% of the isolates. *S*, susceptible; *R*, resistant. * *Staphylococcus* spp. breakpoint.

Antimicrobial Agent	Range (mg/L)	MIC_50_	MIC_90_	S ≤ Breakpoint > R		Resistant (n)	Total (%) R
Ampicillin	0.016–256	1	>256	0.05	2	40	100.0
Erythromycin *	0.016–256	>256	>256	1	2	38	95.0
Gentamicin	0.016–256	4	9	1	1	33	82.5
Levofloxacin	0.002–32	>32	>32	1	1	38	95.0
Linezolid	0.015–256	0.75	1	2	2	1	2.5
Rifampicin	0.002–32	0.016	0.04	0.06	0.5	2	5.0
Tetracycline	0.016–256	2	4	2	2	20	50.0
Vancomycin	0.016–256	0.5	0.75	2	2	0	0.0

**Table 2 antibiotics-09-00404-t002:** MIC values of five *C. urealyticum* whose genomes were sequenced. MICs are expressed in mg/L.

Antimicrobial Agent	VH3073	VH4248	VH4549	VH5757	VH5913
Ampicillin	>256	>256	>256	>256	>256
Erythromycin	>256	>256	>256	>256	>256
Gentamicin	6	6	6	4	4
Levofloxacin	>32	0.19	>32	>32	>32
Linezolid	1	0.38	0.75	1	1
Rifampicin	>32	0.04	>32	0.016	0.016
Tetracycline	1.5	0.25	3	3	6
Vancomycin	0.75	0.38	0.5	0.75	0.75

**Table 3 antibiotics-09-00404-t003:** MIC values (mg/L) of seven *C. urealyticum* and the reference strain DSM 7109 determined in absence and presence of PAβN.

Strain	Tetracycline MIC	Tetracycline + PAβN MIC
VH638	>256	0.016
VH1572	4	<0.016
VH1700	4	0.094
VH2234	32	1
VH4548	4	<0.016
VH5757	3	0.125
VH5913	6	<0.016
DSM 7109	32	32

**Table 4 antibiotics-09-00404-t004:** Primers used in the detection and sequencing of resistance genes. Tª indicates annealing temperatures. Size refers amplicon sizes in base pairs.

Gene	Resistance	DNA Sequence (5′-3′)	Tª	Size	Reference
*ermX*	Erythromycin	AACCATGATTGTGTTTCTGAACG ACCAGGAAGCGGTGCCCT	57 °C	560	[30]
*ermB*	Erythromycin	GAAAAGGTACTCAACCAAATA AGTAACGGTACTTAAATTGTTTAC	52 °C	639	[30]
*mef(A-E)*	Erythromycin	GCAAATGGTGTAGGTAAGACAACT TAAAACAAATGTAGTGTACTA	52 °C	399	[30]
*blaA*	Penicillin	CAGTCTAGCCACTTCGCCAAT TGACTGCACGGATGGAGATGG	55 °C	808	[31]
*gyrA*	Levofloxacin	GCGGCTACGTAAAGTCC CCGCCGGAGCCGTTCAT	60 °C	337	[32]
*tetA*	Tetracycline	TTAGCGTTCGGCGACCTGG GGTGGTCTTGTCTGCCCTCA	60 °C	552	This study
*tetB*	Tetracycline	ACGGTGTTCAACGCCCTGTT AACTGGGTGCCTTCAGGGTC	59 °C	506	This study
*rpoB*	Rifampicin	CTGATCCAGAACCAGGTCCG GACGTACTCCACCACACCAG	55 °C	811	This study
*aac(3)-XI*	Gentamicin	GCGGCTACGTAAAGTCC CCGCCGGAGCCGTTCAT	60 °C	337	[14]

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
