# Peer review of "Antimicrobial Susceptibility and Characterization of Resistance Mechanisms of Corynebacterium urealyticum Clinical Isolates"

_antibiotics, 2020, doi:10.3390/antibiotics9070404_

Round 1
Reviewer 1 Report
Antibiotics-Original article revision: “Antimicrobial susceptibility and characterization of resistance mechanisms of Corynebacterium urealyticum clinical isolates”
The aim of the study was to investigate the genetic relationship and the antimicrobial resistance pattern of 40 C. urealyticum clinical isolates from Spanish hospitals.
General observations
The article highlights interesting findings in the field of the treatment options of infections caused by C. urealyticum, considered a human opportunistic pathogen. In particular, Authors retrospectively analyzed 40 clinical isolates from Spanish hospitals, comparing their genetic characteristics and the resistance pattern, for which literature data are scarce. As a result, 39/40 isolates showed a MDR profile, with high rates of resistance for ampicillin, levofloxacin, erythromycin, gentamicin and tetracycline. Vancomycin, linezolid and rifampicin showed a good antimicrobial activity. Isolates were not clonally related. These data suggest that resistance to antibiotics in C. urealyticum is a cause of concern that should be monitored by continuous surveillance strategies.
I have only some minor points:
- Resistance to linezolid, to my opinion, is one of the most important data of this work, representing a very rare (maybe unique?) finding. Authors should describe (and discuss) the mechanism of resistance to linezolid in the resistant isolate.
- Resistance to gentamicin in your isolates is not clear. 82% of isolates are resistant, but PCR for aac gene were negative. Partial genes were found by WGS in 4/5 isolates. Authors should explain, in a more clear manner, if resistance to gentamicin is due to aac or not. Could you suggest any alternative resistance mechanisms?
- 50% of isolates were resistant to tetracycline, but efflux encoded by tet genes have been excluded. Authors should discuss alternative resistance mechanisms.
- Please, specify in M&M the EUCAST version for interpretation of S-I-R categories. In v. 10.0, breakpoints for gentamicin have not been provided.
- Please, specify in M&M the collection period of clinical isolates.
Author Response
MS ANTIBIOTICS-845545 REVISION
RESPONSES TO REVIEWER 1
Comments and Suggestions for Authors
Antibiotics-Original article revision: “Antimicrobial susceptibility and characterization of resistance mechanisms of Corynebacterium urealyticum clinical isolates”
The aim of the study was to investigate the genetic relationship and the antimicrobial resistance pattern of 40 C. urealyticum clinical isolates from Spanish hospitals.
General observations
The article highlights interesting findings in the field of the treatment options of infections caused by C. urealyticum, considered a human opportunistic pathogen. In particular, Authors retrospectively analyzed 40 clinical isolates from Spanish hospitals, comparing their genetic characteristics and the resistance pattern, for which literature data are scarce. As a result, 39/40 isolates showed a MDR profile, with high rates of resistance for ampicillin, levofloxacin, erythromycin, gentamicin and tetracycline. Vancomycin, linezolid and rifampicin showed a good antimicrobial activity. Isolates were not clonally related. These data suggest that resistance to antibiotics in C. urealyticum is a cause of concern that should be monitored by continuous surveillance strategies.
I have only some minor points:
- Resistance to linezolid, to my opinion, is one of the most important data of this work, representing a very rare (maybe unique?) finding. Authors should describe (and discuss) the mechanism of resistance to linezolid in the resistant isolate.
Unfortunately the linezolid resistant strain genome was not fully sequenced. We have performed alignments of RplC and RplD proteins of the 5 full-sequenced strains (linezolid-susceptible) and we have not found differences. The genome sequencing of the linezolid-resistant strains is envisaged to be done in the future, but not on time for this article.
- Resistance to gentamicin in your isolates is not clear. 82% of isolates are resistant, but PCR for aac gene were negative. Partial genes were found by WGS in 4/5 isolates. Authors should explain, in a more clear manner, if resistance to gentamicin is due to aac or not. Could you suggest any alternative resistance mechanisms?
We hypothesize that low level gentamicin resistance in 4 isolates is due to the presence of the aac(3)-XI gene whereas in the fifth isolate is due to another mechanism.
- 50% of isolates were resistant to tetracycline, but efflux encoded by tet genes have been excluded. Authors should discuss alternative resistance mechanisms.
The tetracycline MICs of 7 tetracycline-resistant C. urealyticum have been tested in presence and absence of the broad-spectrum efflux pump inhibitor Phe-Arg-b-naphthylamide (PAbN). Significative decrease in resistance level in presence of (PAbN) indicates that tetracycline resistance is produced by an efflux mechanism.
These results have been incorporated in RESULTS (L287-297, Table 3), DISCUSSION (L401-403) and M&M (L432-433) sections of the manuscript.
- Please, specify in M&M the EUCAST version for interpretation of S-I-R categories. In v. 10.0, breakpoints for gentamicin have not been provided.
The following paragraph has been introduced in M&M section 4.2 (L434-439):
Clinical categories were established according to the breakpoints for the microdilution susceptibility assay defined by the European Committee on Antimicrobial Susceptibility Testing (EUCAST) (http://www.eucast.org) guidelines [29]. EUCAST defined Corynebacterium spp. specific breakpoints for vancomycin, linezolid, tetracycline, and rifampicin; for ampicillin, gentamicin, and levofloxacin we considered EUCAST PK-PD breakpoints. For erythromycin, Staphylococcus spp. cut-offs defined by EUCAST were used.
- Please, specify in M&M the collection period of clinical isolates.
It has been specified in L416: "during the period 2005-2017".
Reviewer 2 Report
The manuscript entitled "Antimicrobial susceptibility and characterization of resistance mechanisms of Corynebacterium urealyticum clinical isolates" is a well done and interesting paper, pointing out the need of manteining the surveillance of resistance in oportunistic pathogens, especially those that are related with the human microbiota.
But there are some questions.
1.- It is interesting to know which are the resistant profile of the 5 strains sequenced. A table showing all the different resistant patterns and the number of strains in each profile could be appropiated.
2.- In the subtitle 2.3 wrote " Fig 4. The reference strain ATCC 43042...." but in figure 4 this strain did not appear. The Fig 4 showed the strain DSM7109.
3.- Determinants of tetracycline resistant were not found by PCR ( tetA, tetB), so the mechanisms of resistance to tetracycline are not characterized. An aproximation to find efflux mechanisms to tetracycline should be to test the MIC in presence of any Efflux pump inhibitor like reserpine or others.
4.- Characterization of quinolone resistance are not included in RESULTS section, are only found in discussion. Add a phrase or paragraph in results.
5.- Despite linezolid resistance were found only in one strain the authors did not characterise its mechanism of resistance. If this strain belongs to a full sequenced one, it could be interesting to look for any mutation in any of the 16 S RNA gene that corynebacetrium have. Or in any of the ribisomal proteins involved in linezolid resistance as L3 or L4 among others.
6.- lines 334-341 had duplicated information that are already on the table 2.
Author Response
MS ANTIBIOTICS-845545 REVISION
RESPONSES TO REVIEWER 2
REVIEWER 2
Comments and Suggestions for Authors
The manuscript entitled "Antimicrobial susceptibility and characterization of resistance mechanisms of Corynebacterium urealyticum clinical isolates" is a well done and interesting paper, pointing out the need of mantaining the surveillance of resistance in opportunistic pathogens, especially those that are related with the human microbiota.
But there are some questions.
1.- It is interesting to know which are the resistant profile of the 5 strains sequenced. A table showing all the different resistant patterns and the number of strains in each profile could be appropriate.
This table has been introduced: Table 2, L207
2.- In the subtitle 2.3 wrote " Fig 4. The reference strain ATCC 43042...." but in figure 4 this strain did not appear. The Fig 4 showed the strain DSM7109.
The C. urealyticum reference strain is named as strain DSM 7109 in all the manuscript.
3.- Determinants of tetracycline resistant were not found by PCR ( tetA, tetB), so the mechanisms of resistance to tetracycline are not characterized. An aproximation to find efflux mechanisms to tetracycline should be to test the MIC in presence of any Efflux pump inhibitor like reserpine or others.
The tetracycline MICs of 7 tetracycline-resistant C. urealyticum have been tested in presence and absence of the broad-spectrum efflux pump inhibitor Phe-Arg-beta-naphthylamide (PAbN). Significative decrease in resistance level in presence of (PAbN) indicates that tetracycline resistance is produced by an efflux mechanism.
These results have been incorporated in RESULTS (L287-297, Table 3), DISCUSSION (L401-403) and M&M (L432-433) sections of the manuscript.
4.- Characterization of quinolone resistance are not included in RESULTS section, are only found in discussion. Add a phrase or paragraph in results.
A paragraph describing the levofloxacin resistance mechanism has been introduced in the RESULTS section 2.2. (L217-225).
5.- Despite linezolid resistance were found only in one strain the authors did not characterise its mechanism of resistance. If this strain belongs to a full sequenced one, it could be interesting to look for any mutation in any of the 16 S RNA gene that corynebacetrium have. Or in any of the ribosomal proteins involved in linezolid resistance as L3 or L4 among others.
Unfortunately the linezolid resistant strain does not belong to the full sequenced group. We have performed alignments of RplC and RplD proteins of the 5 full-sequenced strains (linezolid-susceptible) and we have not found differences. The genome sequencing of the linezolid-resistant strain is envisaged to be done in the future, but not on time for this article.
6.- lines 334-341 had duplicated information that are already on the table 2.
This duplicated information in M&M section 4.3. has been eliminated.